# UHPLC-ESI-OT-MS Phenolics Profiling, Free Radical Scavenging, Antibacterial and Nematicidal Activities of “Yellow-Brown Resins” from *Larrea* spp.

**DOI:** 10.3390/antiox10020185

**Published:** 2021-01-28

**Authors:** Jessica Gómez, Mario J. Simirgiotis, Sofía Manrique, Mauricio Piñeiro, Beatriz Lima, Jorge Bórquez, Gabriela E. Feresin, Alejandro Tapia

**Affiliations:** 1Instituto de Biotecnología-Instituto de Ciencias Básicas, Universidad Nacional de San Juan, Av. Libertador General San Martín 1109 (O), San Juan CP 5400, Argentina; jesicagomez674@gmail.com (J.G.); manriquesofia2@gmail.com (S.M.); mauridpg@gmail.com (M.P.); blima@unsj.edu.ar (B.L.); gferesin@unsj.edu.ar (G.E.F.); 2CONICET (Consejo Nacional de Ciencia y Tecnología), CABA, Buenos Aires C1405DJR, Argentina; 3Instituto de Farmacia, Facultad de Ciencias, Campus Isla Teja, Universidad Austral de Chile, Valdivia 5090000, Chile; 4Center for Interdisciplinary Studies on the Nervous System (CISNe), Universidad Austral de Chile, Valdivia 5090000, Chile; 5Laboratorio de Productos Naturales Depto. de Química, Facultad de Ciencias, Universidad de Antofagasta, Av. Coloso S-N, Antofagasta 1240000, Chile; jorge.borquez@uantof.cl

**Keywords:** dipping in dichloromethane, biomolecules of pharmacological interest, *Larrea divaricata*, *L. nitida*, lignans

## Abstract

This research was designed to investigate the metabolite profiling, phenolics and flavonoids content and the potential antioxidant, antibacterial and nematicidal activities of “yellow-brown resins” from *Larrea divaricata* Cav (LdRe) and *L. nitida* Cav (LnRe). Metabolite profiling was obtained using an ultrahigh resolution liquid chromatography orbitrap MS analysis (UHPLC-ESI-OT-MS). The antioxidant properties were screened by four methods: 2,2-diphenyl-1-picrylhydrazyl assay (DPPH), trolox equivalent antioxidant activity assay (TEAC), ferric-reducing antioxidant power assay (FRAP) and lipid peroxidation in erythrocytes (LP). The antibacterial activity was evaluated according to the Clinical and Laboratory Standards Institute (CLSI) guidelines. In addition, the potential combinatory effect was analyzed with the fractional inhibitory concentration index (FICI) values using the checkerboard design. The nematicidal activity was carried out according to a standardized protocol. LdRe and LnRe showed a strong capture of the DPPH radical withvalues around 8.4 µg resin/mL; FRAP (1.69–1.94 mgTE/ g resin), TEAC (1.08–1.09 mgTE/g resin) and LP (81–82% at 100 µg of resin/mL) assays. A strong antimicrobial activity was displayed by both resins against methicillin-sensitive *Staphylococcus aureus* ATCC 25923(MSSA) and methicillin-resistant *S. aureus* ATCC 43300(MRSA) (MICs = 16–32 µg resin/mL). Additionally, the combination of LdRe or LnRe with the antibiotic cefotaxime showed an indifferent effect (FICI values = 1–1.25), however, this combinationcould be a potential strategy to reduce the drug doses, and in this way can be a potential alternative to reduce bacterial resistance. On the other hand, the resins showed a scarce nematicidal potential toward J2 *Meloidogyne incognita*; an important nematode infecting horticultural crops. Phenolics compounds were identified by UHPLC-PDA-OT-MS analysis, updating the knowledge on the chemical profile of these species. These results, together with the high content of quantified phenolics and flavonoids, allow the phenolics-enriched resins of these two *Larrea* species to be considered as a promising sustainable source of compounds of pharmacological interest.

## 1. Introduction

The resins are naturally secreted by species of the families Asteraceae, Burseraceae, Fabaceae, Pinaceae and Zygophyllaceae. Their chemical profiles are characterized by phenolic compounds, terpenes, flavonoids, chalcones, lignans and fats, some of which display protective functions against natural predators and pathogens of these species [1]. Exudates and resins from some South American plants have been previously reported as sources of potential antimicrobial and antioxidant compounds [2,3,4,5]. Argentina and Mexico are the main habitat of resinous plant species belonging to the genera *Larrea* (Zygophyllaceae). From this genus, four species are widely distributed in Argentina (*Larrea ameghinoi*, *L. cuneifolia*, *L. divaricata* and *L. nitida*); two of the species also inhabit Chile (*L. divaricata* and *L. nitida*) and two species grow in Mexico (*L. mexicana* and *L. tridentata)* [6,7]. The species of the genus *Larrea* in Argentina (*Larrea ameghinoi*, *L. cuneifolia*, *L. divaricata* and *L. nitida*), vernacular name “jarillas” are used extensively in traditional medicine in Argentina and in Andean communities for the treatment of injuries and bruises, and a good disinfectant of wounds, repellent of insects, for roof construction in rural areas and as a vegetable fuel for cooking food. These uses are also shared with the resinous species *Zuccagnia punctata*, commonly called “jarilla macho” [5]. Extracts and infusions of aerial parts of most species of this genus have displayed several biological activities such as antimicrobial, antioxidant and inhibitory agents of several enzymes, anti-inflammatory, antitumor and others [8,9,10,11,12,13,14,15,16,17,18]. However, the chemical characterization of the *Larrea* genus resins, characterized by their high content of phenolics compounds, is still scarce, and their potential antioxidant, antibacterial and nematicidal activities.

In previous reports, several medicinal plants and native fruits growing in Argentina and Chile have been analyzed combining full mass spectra and MSn experiments from quadrupole orbitrap spectrometry (Q-OT-MS), significantly updating the chemical composition and their biological activities, in most of the species investigated. A considerable number of these previous reports have given scientific support to the use or consumption in traditional medicine in both countries [19,20,21,22,23,24,25].

The main goals and novelty of this work are the antioxidant, antibacterial and nematicidal effects complemented with the full metabolome polyphenolic profile using a hybrid high-resolution mass spectrometer of the resins from the medicinal plants, *L. divaricata* and *L. nitida*, supporting the recognized medicinal properties of this plant as a sustainable way of biomolecules of pharmacological interest.

## 2. Materials and Methods

### 2.1. Chemicals

Ultra-pure water (<5 µg/L TOC, (total organic carbon)) was obtained from a water purification system Arium 126 61316-RO, plus an Arium 611 UV unit (Sartorius, Goettingen, Germany). Methanol (HPLC grade) and formic acid (puriss. p.a. for mass spectrometry) from J. T. Baker (Phillipsburg, NJ, USA) were obtained. Dichloromethane (HPLC grade) were from Merck (Santiago, Chile). Commercial Folin-Ciocalteu(FC) reagent, 2,2-diphenyl-1-picrylhydrazyl (DPPH), ferric chloride hexahydrate, 2,4,6-tris(2-pyridyl)-s-triazine, trolox, quercetin, gallic acid and DMSO were purchased from Sigma-Aldrich Chem. Co. (St Louis, MO, USA). Nordihydroguaiaretic acid (NDGA), 3′ methyl-nordihydroguaiaretic acid (MNDGA), LnRe and characterized by analysis of their spectroscopic data (^1^H and ^13^C NMR, MS) were used as standards [26].

### 2.2. Plant Material

Three representative and independent samples of 2 kg of aerial parts (stems with leaves and flowers) for each species were collected in December 2015, in the Iglesia district, province of San Juan (Argentina) at an altitude of 1800 m (*L. divaricata*) and 2700 m (*L. nitida*) above sea level. Each sample was collected from ten plants. The two plants belonging to the *Zygophyllaceae* genera were classified and a voucher specimen was deposited in the Laboratory of Natural Products of the University of San Juan (Argentina) under the following voucher number IBT-Ld-1and IBT-Ln-1.

### 2.3. Extraction of Yellow-Brown Resin

*L. divaricata* Cav (LdRe) and *L. nitida* Cav (LnRe) resins were obtained by dipping the fresh plant (100 g) in cold CH_2_Cl_2_(DCM, 1000 mL) at room temperature (30 °C) for 60 s. Each DCM extract was concentrated under pressure at 30 °C to give a following semisolid yellow-brown residue for each *Larrea* resin: LdRe resin12 g, (12% *w/w* yield) and LnRe resin 9.5 g, (9.5% *w/w* yield). The extraction procedure was done three times.

### 2.4. UHPLC-DAD-MS Instrument and Chromatographic Conditions

A Thermo Scientific Dionex Ultimate 3000 UHPLC system, hyphenated with a Thermo high resolution Q-Exactive focus mass spectrometer (Thermo, Bremen, Germany) was used for the analysis. The chromatographic system was coupled to the MS with a heated electrospray ionization source II (HESI II). Nitrogen (purity > 99.999%) obtained from a Genius NM32LA nitrogen generator (Peak Scientific, Billerica, MA, USA) was employed to produce MS fragmentation. Mass calibration for the Orbitrap spectrometer and HESI parameters were explained in detail and precisely in previous reports [5,24].

Solvent delivery was performed at 1 mL/min using ultrapure water supplemented with 1% formic acid (A) and acetonitrile with 1% acid formic (B) and a program starting with 5% B at zero time, then maintained 5% B for 5 min, then changing to 30% B within 10 min, then maintaining 30% B for 15 min, then going to 70% B for 5 min, then maintaining 70% B for 10 min, and finally returning to 5% B in 10 min and keeping this condition for twelve additional minutes to achieve column stabilization before the next injection of 20 µL [5,24].

### 2.5. Determination of Total Phenolic (TP) and Flavonoid (F) Content

The total content of phenolic compounds and flavonoids was determined by means of the Folin–Ciocalteu and AlCl_3_ tests respectively [5,24]. The evaluated concentrations of the resins were 1 mg/mL. The results were obtained by using standards (gallic acid and quercetin) and were expressed as equivalent in milligrams to these (mg GAE/g *Larrea* resin for phenolics and mg QE/g *Larrea* resin for flavonoids).

### 2.6. Antioxidant Activity

#### 2.6.1. DPPH Scavenging Activity

The radical scavenging capacity of both resins was assessed using the DPPH assay, which was explained in detail and precisely for reproducibility in previous reports [5,24]. The resins and reference standards were analyzed in the concentration range of 1 and 100 µg/mL. The values obtained were expressed as mean EC_50_ ± SD in µg resin/mL.

#### 2.6.2. Ferric-Reducing Antioxidant Power Assay (FRAP)

The reducing power of the *Larrea* resins was evaluated using the FRAP test, following the protocols already reported explained in detail for correct reproducibility [5,24]. Both resins were tested at 1 mg/mL. In addition, a Trolox calibration curve (0–1 mmol/L) was used. The results obtained were reported as milligrams Trolox equivalent by/grams of resin (mg TE/g resin).

#### 2.6.3. Trolox Equivalent Antioxidant Activity (TEAC) Assay

The assay was carried out in the microplate protocol, which has been reported in detailed form for their correct reproducibility in previous reports [5,24]. The LdRe and LnRe resins were dissolved in methanol and mixed with 200 µL of ABTS, reading the absorbance at 734 nm after 4 min. Results from a calibration curve constructed with Trolox (reference compound, 0–1 mmol/L) are expressed as equivalent milligrams Trolox by grams resin (mg TE/g resin).

#### 2.6.4. Lipid Peroxidation in Human Erythrocytes

The protocol to lipid peroxidation in human erythrocytes (LP) assay was explained in detail and precisely for reproducibility in previous reports [5,24]. Blood samples were obtained from healthy volunteer donors, and extracted by doctors in Biochemistry belonging to our Research Institute. LdRe and LdRe were assayed 100 and 250 µg of *Larrea* resin/mL, while the reference compound catechin was at 100 µg resin/mL. The results are expressed as a percentage of inhibition of LP in µg of resin/mL. Healthy volunteer donors gave their informed consent for inclusion before participating in the study. The protocol assay was conducted in accordance with the Declaration of Helsinki, and is included in the approvedproject by CICIT-CA-UNSJ-Argentina (Projectcode 80020190100277SJ, Resolución N 0591-20-R-UNSJ) referred to the ethical and environmental safeguard, and to preserve the hygiene and safety conditions in the activities to be carried out in laboratories.

### 2.7. Nematicidal Activity

The nematicidal effects were determined by the procedure previously described [27]. A pure population of *Meloidogyne incognita* previously identified was collected from infected *Solanum lycopersicum* fields from San Juan province, Argentina. Infected roots were washed gently and sterilized with 1% NaOCl for 4 min. Under stereoscopic microscope at 1.6 × egg masses hand-picked. They were incubated in a growth chamber at 27 ± 1 °C. After hatching, second stage juveniles (J2) were collected up to 3 days old for the assays. For the screening, 50 mg of *Larrea* resins were dissolved in 98.7 mL distilled water, 1 mL MeOH and 0.3 mL Tween 20. These solutions were filtered on filter paper and the final concentration was calculated for each one and constituted as the undiluted solution. The final concentrations for LdRe was 0.023% *w/v* and for LnRe was 0.025% *w/v*, each solution were testing undiluted (1), dilution 1:1 (2) and dilution 1:4 (3) *v/v*. Thirty J2 were incubated in glass Petri dishes of 50 mm ø with 10 mL of *Larrea* resin and kept in the dark at 28 ± 1 °C. Regarding the negative control, 10 mL of purified water was used. At 72 h under stereoscopic microscope, the vitality of the nematodes was corroborated. Five replicates were made for each treatment and for the negative control. J2 were considered as dead if their bodies were straight with no movement even if physically stimulated with a fine needle. Data were expressed as percentage mortality of J2 and to calculate the correction % by natural mortality in control, Schneider-Orelli formula was applied: Corrected % = (mortality % in treatment-mortality % in control))/((100-mortality % in control)) × 100, and they were analyzed by ANOVA and LSD Fisher α = 0.05 to determine the statistical differences between the means, through InfoStat program (v.2018).

### 2.8. Antibacterial Activity

#### 2.8.1. Microorganisms

Gram-positive strains: methicillin-sensitive *Staphylococcus aureus* ATCC 29213 (MSSA), methicillin-resistant *Staphylococcus aureus* ATCC 43300 (MRSA), *Staphylococcus aureus*-MQ5097, *Streptococcus pyogenes*-MQ4 and *Streptococcus agalactiae* and Gram-negative strains: *Escherichia coli* ATCC 25922, *E.coli*-MQ11009, *E. coli*-MQ11068, *E. coli*-MQ11062 and *E. coli*-MQ586, from the American Type Culture Collection(ATCC, Rockville, MD, USA) and clinical isolates provided by the Microbiology Laboratory of the Public Dr. Marcial Quiroga, Hospital, San Juan, Argentina (MQ) were used.

#### 2.8.2. Antibacterial Activity of *Larrea* Resins

Minimum inhibitory concentration (MIC) of resins and reference antibiotics Cefotaxime (Argentia^®^, Buenos Aires, Argentina) and Imipenem-cilastatin (Imipecil^®^) were carried out by broth microdilution techniques using Mueller Hinton medium, according to Clinical and Laboratory Standards Institute (CLSI) [28]. The LdRe and LnRe resins were tested in triplicate between 0.98 and 250 µg/mL. Microorganism suspensions were adjusted in a spectrophotometer with sterile physiological solutions to give a final organism density of the 0.5 McFarland scale (5 × 10^5^ CFU/mL) A volume of 100 µL of inoculum suspension was added to each well with the exception of the sterility control. The samples were diluted in DMSO in the assay (≤1%). The absorbances at 620 nm were determined in a Multiskan FC Microplate Photometer (Thermo Scientific, Waltham, MA, USA). Minimum inhibitory concentration (MIC in µg/mL) was defined as the lowest compounds/extracts concentration showing no visible bacterial growth after incubation time. The minimum bactericidal concentration (MBC in µg/mL) of *Larrea* resins and reference compounds was determined as follows: After determining the MIC, an aliquot of 5 μL sample was withdrawn from each clear well of the microtiter tray and plated onto a agar plate. Inoculated plates were incubated at 37 °C, and MBCs were recorded after 24 h. The MBC was defined as the lowest concentration of each compound that resulted in total inhibition of visible growth in these plates.

#### 2.8.3. Antibacterial Combinatory Effect between *Larrea* Resins with Cefotaxime by the Checkerboard Design

The potential antibacterial combinatory effect (synergism) between LdRe, LnRe and reference antibiotic cefotaxime (Cef) was evaluated against methicillin-sensitive *Staphylococcus aureus* ATCC 25923 (MSSA) and methicillin-resistant *S. aureus* ATCC 43300 (MRSA), using the checkerboard design [29]. Briefly, LdRe and LnRe were diluted two-fold in the vertical orientation, whereas antibacterial drugs were diluted in the horizontal one. Their respective concentrations were 4, 2, 1/2, 1/4 and 1/8 × MIC, which were selected based on MIC values previously determined. In each well (microliter plate) was added an inoculum of 5 × 10^5^ (CFU/mL) and incubated at 37 °C (24 h). The checkerboard design are showed in Appendix A was as follows:

For each combination, the fractional inhibitory concentration (FIC) was calculated as follows:(1)FICLarrea resin=MICLarrea in comb.MICLarrea resin alone
(2)FICCef=MICCef in comb.MICCef alone

Additionally, the fractional inhibitory concentration index (FICI) was determined by following equation:(3)$$FICI=FIC_{Larrea\;resin}+FIC_{Cef}$$

The values obtained were interpreted as follow: synergy effect (FICI ≤ 0.5), no interaction or indifference (FICI > 0.5–4.0), and antagonism (FICI > 4) according to Odds, (2003) [30].

#### 2.8.4. Dose Reduction Index (DRI)

The dose reduction index (DRI) determines how manyfolds the dose of an antibiotic in a combination may be reduced at a given effect level compared with the doses of the antibiotic alone. A greater DRI indicates a greater dose reduction for a given effect level [31]. The DRI value for cefotaxime was calculated as follows: (4)DRI=MICCef aloneMICCef in comb.

#### 2.8.5. Isobolograms

The potential antibacterial combinatory effect between LdRe and LnRe and cefotaxime antibiotic are showed in normalized Isobolograms, which depicts the results of the checkerboard design [32]. In the Isobolograms, the alone MIC value of *Larrea* resins (LdRe o LnRe) is represented on the x-axis, and the alone MIC of the cefotaxime on the y-axis. The line of no interaction (line of indifference) represents the line connecting these two points. The combinations below this line are a synergistic effect (FICI ≤ 0.5), while combinations above represent an antagonism effect (FICI > 4) [30].

### 2.9. Statistical Analysis

All determinations in total phenolic (TP), flavonoid (FT) and antioxidant assays were made in triplicate using a Multiskan FC Microplate Photometer (Thermo Scientific, Waltham, MA, USA). The values from triplicates are reported as the mean ± standard deviation (SD). In addition, the Duncan’s test of the InfoStat program (2016 edition, National University of Córdoba, Argentina) was used to determine statistical differences (*p* < 0.05) between the tests used.

## 3. Results

### 3.1. UHPLC-OT Analysis of LdRe and LnRe

The use of full scan mass spectra, using base peaks chromatograms and fragmentation experiments were very useful for the identification of a significant number of flavonoids, epoxylignans and cyclolignans, together with some isomer compounds characteristics of these bioactive plants since the orbital trap provided high-resolution and accurate mass product ion spectra for untargeted analyses within a single run.

Combining full MS spectra and some diagnostic MSn experiments, forty-one compounds were detected in LdRe and LnRe resins, by ultrahigh resolution liquid chromatography Q-orbitrap MS analysis (UHPLC-PDA-Q-OT-MS). From them, the detected and characterized compounds were tentatively proposed as flavonoids, lignans and their derivatives. The generation of molecular formulas was performed using high resolution accurate mass analysis (HRAM) and matching with the isotopic pattern. In this work only negative mode of detection was used. Electrospray negative mode with energy of “0” or “5” EV is the most abundantly used method to detect phenolics. Compounds with a phenolics OH easily lose the proton in electrospray ionization, giving very good and diagnostic parent ions and fragments. Finally, analyses were confirmed using MS/MS data and comparing the fragments found with the literature data. The complete metabolome identification is showed below involving Figure 1, Figure 2 and Figure 3, Table 1; and Appendix A.

Forty-one phenolics compounds were detected and tentatively identified by ultrahigh resolution liquid chromatography orbitrap MS analysis in LdRe and LnRe resins. Of these, one was a phenolic compound (**1**), eight wereflavonoids (**2, 3, 5, 7 ,8, 9, 15** and **22**) and twenty-nine tentatively proposed as lignans (**4, 6, 10, 11, 12, 13, 16,17, 18, 19, 20, 21, 23, 24, 25, 26, 27, 28, 29, 30, 31, 32, 33, 34, 35, 36, 37, 38, 39** and **40**). From them, a significant number were assigned as NDGA and its derivatives, supported by molecular mass, fragmentation patterns, bibliographic references and databases as SciFinder, MassBank of North America (MoNA) and UHPLC-MS internal library. Finally, peak **41** was assigned as unknown. Peak **4, 13, 18, 19** and **20** were identified as meso-(rel 7S,8S,7′R,8′R)-3,4,3′,4′-tetrahydroxy-7,7′-epoxylignan; nordihydroguaiaretic acid (NDGA); (7S,8S,7′R,8′R)- 3,3′,4′-trihydroxy-4-methoxy-7,7′-epoxylignan; (4-[4-(4-hydroxy-phenyl)- 2,3-dimethyl-butyl]-benzene-1,2-diol) and 3′ methyl-nordihydroguaiaretic acid (MNDGA) respectively by spiking experiments with available standards previously isolated [26]. In a similar way peak **21** was proposed as a 4′-MNDGA, an isomer of **20**, supported by their identically measured accurate mass and major diagnostic MS ions [26].

The compounds **11**, **14**, **31** and **33** were tentatively identified as isomers of NDGA (13), from them, in the compounds **31** and **33**; the parent ion (301.14429) produced the same major diagnostic MS ions at *m/z* 174.95743 and *m/z* 122.03708 confirming these proposed isomers. A synthetic route to the naturally occurring nordihydroguaiaretic acid and its non-meso isomer starting from the commercially available (3,4-dimethoxyphenyl) acetone has been reported [33]. On the other hand, peaks **26** with a at *m/z* 329.17551 was tentatively identified as meso-dihydroguaiaretic acid [34,35]; while [M−H]^−^ ion peak **28** with a [M−H]^−^ ion at *m/z* 281.11813 was tentatively identified as reduced NDGA derivative (Appendix A).

Peaks **6** and **10** with a [M−H]^−^ ion at *m/z* 299.12872 were proposed as 3,4,3′,4′-tetrahydroxy 6,7′-cyclolignan isomers. Peak **12** and **16** with a [M−H]^−^ ion at *m/z* 313.14432 were assigned to norisoguaicin and its isomer, both compounds and other lignans have been previously isolated from *L. divaricata*, *L. nitida* and *L. tridentate* [34,35,36,37,38,39].

Peak **17** was proposed as trihydroxy-6,7´cyclolignan with a [M−H]^−^ ion at *m/z* 283.0611 [40]. Fourteen peaks including **23, 24, 25, 27, 29, 30, 32** and **34**–**40** were proposed as NDGA derivatives, showing all of them as major diagnostic MS ions between 301.14420 and 301.14429 Daltons. From them, peaks **27** and **30**, both with a [M−H]^−^ ion at *m/z* 371.18595 were tentatively identified as dihydroguaiaretic acid acetates, (4′-O acetate and 4-O acetate respectively), which were isolated previously from *L tridentata* extracts [38,40,41]. Peaks **32** and **34** were tentatively proposed as the tri-methoxylated and acetylated isomers: 3″,4″,4′-trymethyl NDGA3′-acetate and 3″,4″,3′-trymethyl NDGA4′-acetate, with [M−H]^−^ ions at *m/z* 385.20175 and 385.20172 respectively, this is in concordance with tri-methoxylated NDGA derivatives previously isolated from the same source [38,40,41]. Peaks **38** and **40** producing the same [M−H]^−^ ion at *m/z* 399.21738 and peak 39 with a [M−H]^−^ ion at *m/z* 399.21729 yielded the same mayor diagnostic MS ion at *m/z* 301.14429, thorough losses of the substituent’s and thus the three compounds were proposed as MNDGA diacetate derivatives (3′Methyl NDGA 3″, 4′-diacetate, 4′Methyl NDGA 3′, 4″-diacetate and 3′Methyl NDGA 4″, 4′-diacetate, respectively). According to our knowledge, the isolation and identification of phenolics, flavonoids and lignans from extracts of *L. divaricata* and *L. nitida* have been reported in some previous works. From *L. nitida* collected in Chile, norisoguaicin16, nordihydroguaiaretic acid (**13**, NDGA) and ferulic acid have previously been reported [36]. Additionally, the antifungal bioassay-guided isolation of *L. nitida* resin (LnRe) led to the isolation of **4, 13, 16, 18, 20** and **21** [26]. Recently, from an ethanol-water extract of the species *L. divaricata*, *L. nitida* and *L. cuneifolia*, through a study by HPLC-MS in the negative mode, twelve phenolics compounds were characterized, including, quercetin-hexoside, quercetin-methyl ether hexoside and lignans (**4, 6, 13, 17, 18, 19, 20** and **21**) [39]. On the other hand, previous phytochemical investigation of the phenolics from *L. divaricata* led to the isolation and characterization of the lignans nor-dihydroguaiaretic acid (**13**), dihydroguaiaretic acid (**26**), norisoguaiacin (**16**), 3′-demethoxyisoguaiacin and several flavonoids, including kaempferol 3-methyl ether (isokaempferide), isorhamnetin (**3**) and luteolin 3′-methyl ether [42]. In this report, forty-one compounds were detected in LnRe resin by ultrahigh resolution liquid chromatography orbitrap MS analysis (UHPLC-PDA-OT-MS). From them, as far as we know, one phenolic compound (**1**), eight flavonoids (**2, 3, 5, 7, 8, 9, 15** and **22**), and nineteen lignans (**14, 23, 24, 25, 26, 27, 28, 29, 30, 31, 32, 33, 34, 35, 36, 37, 38, 39** and **40**) were reported here for the first time in these species, significantly updating the chemical composition of these interesting native plants.

### 3.2. Total Phenolics and Flavonoids Content; Antioxidant, Nematicidal and Antibacterial Activities

The yellow-brown resins of the two *Larrea* species showed a strong free radical scavenging in the DPPH assay, with EC50 values around 8.4 µg resin/mL (Table 2). These values were comparable to the activity shown by recognized antioxidant compounds such as catechin or quercetin. In the same way, the yellow-brown resins exhibited a strong effect in FRAP and TEAC trials. On the other hand, yellow-brown resins of the two *Larrea* presented a high content of TP, highlighting LnRe collected in the Andean mountains over 2500 m above sea level with a value of 459 mg GAE/g LnRe, of which approximately 8.8 percent correspond to flavonoids (40.8 mg QE/g LnRe) (Table 2). Additionally, both resins exhibited a strong inhibition of lipoperoxidation with LP values between 81 and 82% at 100 µg resin/mL compared to catechin, which showed 70% of LP at the same concentration [24]. Other Andean medicinal plants collected in the Andean region of the province of San Juan, located in the central west of Argentina, also showed strong antioxidant properties and high contents of phenolics compounds, which were characterized mainly by flavonoids, chalcones and phenolics of diverse structures [3,5,21].

*Meloidogyne* spp., which causes root-knot diseases in plants, is the most economically important plant-parasitic nematodes worldwide. Due to the harmful effects on humans and the environment, the chemical compounds that have been used, over decades, for the effective control of nematodes, have been losing application fields and opening the way to the need for new nematicides, friendly to the environment. Biopesticides and specifically bionematicides of botanical origin constitute a desirable component of pest management technology and practices [43]. The nematicidal activity of LnRe and LdRe against J2 *M. incognita* is shown in Table 3. Both resins displayed a less nematicidal activity at 72 h to three concentrations assayed.

Regarding the antibacterial activity, LdRe and LdRe showed a very strong activity against MSSA and MRSAwith a MIC values = 16 µg/mL (Table 4). The antimicrobial activity of medicinal plants is considered very interesting with MICs values < 100 μg/ mL for extracts and MIC values <10 μg/ mL for isolated compounds [44]. Additionally, LdRe showed activity against *Streptococcus pyogenes*-MQ4and LnRe was active against *Streptococcus agalactiae* (MIC = 62.5 μg/mL). In respect to Gram (−) bacteria, *E. coli*-MQ586 was more sensitive to both resins (MIC = 62.5 µg/mL), while these displayed a moderated antibacterial activity against *E. coli* ATCC 25922, *E. coli* MQ-11009, *E. coli* MQ 11068 and *E. coli* MQ-11062 (MICs values between 250 and 500 µg/mL). In a previous report, 3′methyl-nordihydroguaiaretic acid (MNDGA)**20**, nordihydroguaiaretic acid (NDGA) **13** and a NDGA derivative **19**, displayed strong activity against *Trichophyton mentagrophytes*, *T. rubrum* and *Microsporum gypseum* (MICs between 15.6 and 31.25 µg/mL). In the same study, the lignans **20** and **13** showed activities against clinical isolates of *Candidas* spp., *Cryptococcus* spp., *T. rubrum* and *T. mentagrophytes* (MICs and MFCs between 31.25 and 62.5 µg/mL).

On the other hand, the combined use of plant extracts with commercial antibiotics is an alternative against microbial infections [45,46]. The potential of *Larrea* resins in combination with cefotaxime for the development of a new antimicrobial source useful for the treatment of infections associated to MSSA and MRSA, was evaluated using the checkerboard design (Appendix A). The results are showed in Table 5 and Figure 4. The combination of LdRe or LnRe with reference antibiotic cefotaxime showed an indifferent effect with FICI values = 1–1.25 (Equations (1) and (2). In addition, a moderated decrease of the individual MIC value of the commercial drug cefotaxime was observed (DRI = 2–4), as indicative of a greater adjuvant potential capacity for a given effect level, which may be due mainly to the polyphenols present in the *Larrea* resins [47,48]. This justifies the search for alternative therapies as the combination of commercial drugs and natural products.

The results represented in the isobologram (Figure 4) agreed with those obtained by the checkerboard analysis. The additive effect was found to both species tested (MSSA and MRSA) due to that the points the experimental combinations are below the line of indifference. Therefore, the combination between resins (LdRe and LnRe) could be an excellent strategy to reduce the drug doses, and thus, achieve the resistance bacteria.

The broad spectrum of biological activities such as antimicrobials, antioxidants, inhibitors of several enzymes, anti-inflammatory, antitumor and others, whichhave been reported for extracts and infusions of aerial parts of most species of this genus, have been regularly associated with the presence and content of phenolics or lignans and their derivatives [6,7,8,9,10,11,12]. In a previous work, the bioassay-guided isolation of *L. nitida* resin (LnRe) led to the isolation of 3′ methyl-nordihydroguaiaretic acid (MNDGA)**20** and nordihydroguaiaretic acid (NDGA)**13** and compound **19**(4-[4-(4-hydroxy-phenyl)-2,3-dimethyl- butyl]-benzene-1,2-diol) as the compounds mainly responsible for the antifungal activity. Additionally, two epoxylignans: meso-(rel 7S,8S,7′R,8′R)-3,4,3′,4′-tetrahydroxy- 7,7′-epoxylignan and compound 6 (7S,8S,7′S,8′S)-3,3′,4′-trihy droxy-4-methoxy-7,7′-epoxylignan were isolated [26]. Recently the compounds **13**, **19** and **20** have been reported as marker compounds of synergistic mutual potentiation of antifungal activity of *Zuccagnia punctata* Cav. and *L. nitida* Cav. extracts in clinical isolates of *Candida albicans* and *Candida glabrata*. Additionally, in this previous work carried out in an Andean population in the center-west of Argentina, three marker compounds (**13**, **19** and **20**) for *L nitida* were quantified over a year (February, May, September and November), highlighting their higher production in coincidence with the flowering period of the species, late spring and early summer (November and December) [11]. From the forty-one phenolics compounds detected and tentatively identified by UHPLC-ESI-OT-MS-MS in LdRe and LnRe resins, twenty-nine are lignans, mainly simple bisphenyl lignans and tricyclic lignans (cyclolignans and epoxylignans). The species belonging to the *Larrea* genus are rich in simple bisphenyl lignans and tricyclic lignans known as cyclolignans; those compounds support the pharmacological activities of extracts of this species as antiherpes, antioxidant, antifungal and anti-inflammatory. In the last decade, the strong activity of some lignans against the human immunodeficiency virus, human papilloma virus and cancer, their use in ameliorating neurodegenerative diseases and symptoms of aging has been reported. Additionally, molecular mechanisms underlying the antiviral and anticancer activities have been elucidated [40].

## 4. Conclusions

In this report, forty-one compounds were detected in LnRe resin by ultrahigh resolution liquid chromatography orbitrap MS analysis (UHPLC-PDA-OT-MS). From them, as far as we know, one phenolic compound, eight flavonoids and nineteen lignans are reported for the first time to this species, significantly updating the chemical composition of this interesting species. The results here reported support the recognized medicinal properties of these two recognized South American medicinal plants and on the other hand highlights them as a sustainable way of biomolecules of pharmacological interest. Additionally, relevant biological properties can be expected for these two species supported by the content and structural diversity mainly on the detection of bioactive lignans. The yellow-brown resins of the two *Larrea* showed a high content of TP, highlighting LnRe collected in the Andean mountains. This allows us to conclude that the resins from those wild species *L divaricata* and *L nitida* growing in Argentina represent a rich and sustainable source of antioxidants, mainly lignans. Additionally, the antibacterial activity of LdRe and LdRe showed a very strong activity against MSSA and MRSA. The combination between resins (LdRe and LnRe) with cefotaxime could be a potential strategy to reduce the drug doses, and in this way can be a potential alternative to reduce bacterial resistance. The new trend towards the use of natural extracts as pharmaceuticals rather than pure drugs opens a pathway for the development of a phytomedicine derived from resins of *Larrea*. This could be a promising pathway that needs to be intensively explored and developed in the near future.

## Figures and Tables

**Figure 1 antioxidants-10-00185-f001:**
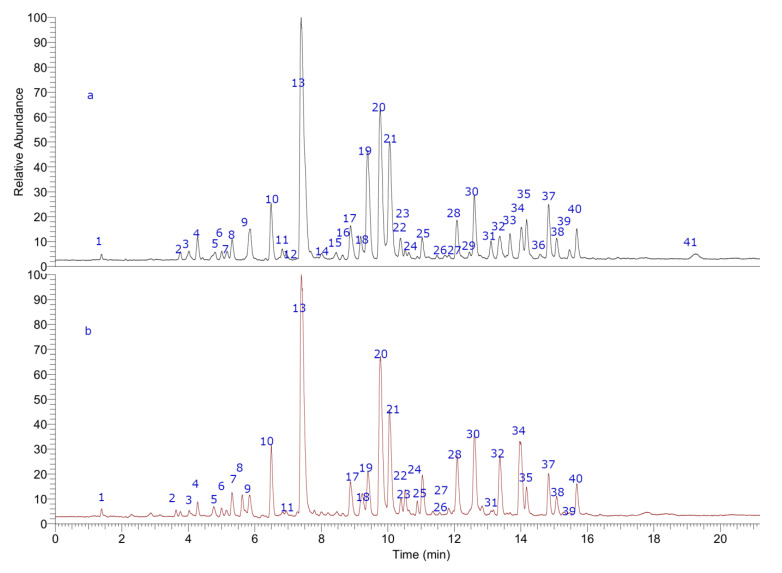
UHPLC-MS (total ion current) chromatograms of LnRe (**a**) and LdRe (**b**) resins.

**Figure 2 antioxidants-10-00185-f002:**
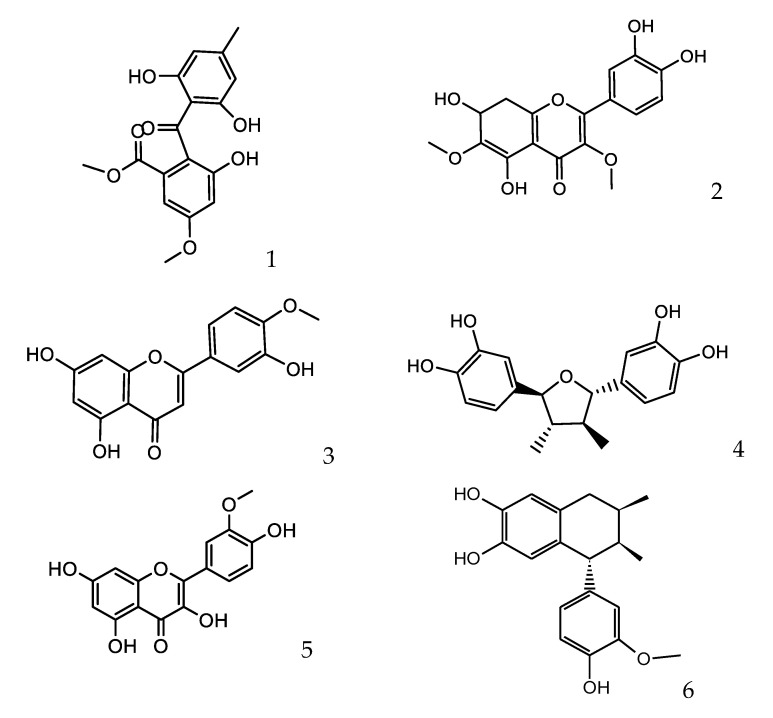
Flavonoids, epoxylignans and cyclolignans characterized in LnRe and LdRe resins.

**Figure 3 antioxidants-10-00185-f003:**
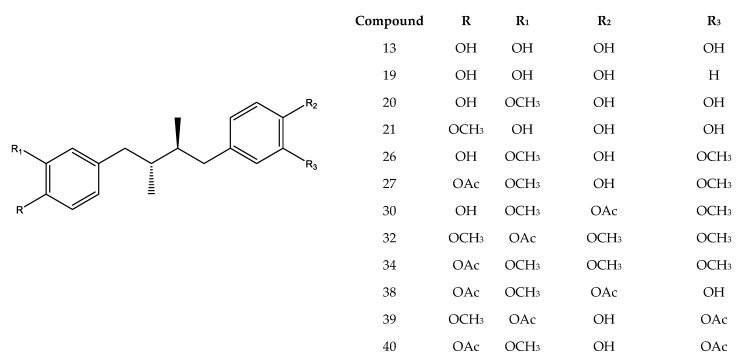
Main NDGA and MNDGA lignans derivatives characterized in LnRe and LdRe resins.

**Figure 4 antioxidants-10-00185-f004:**
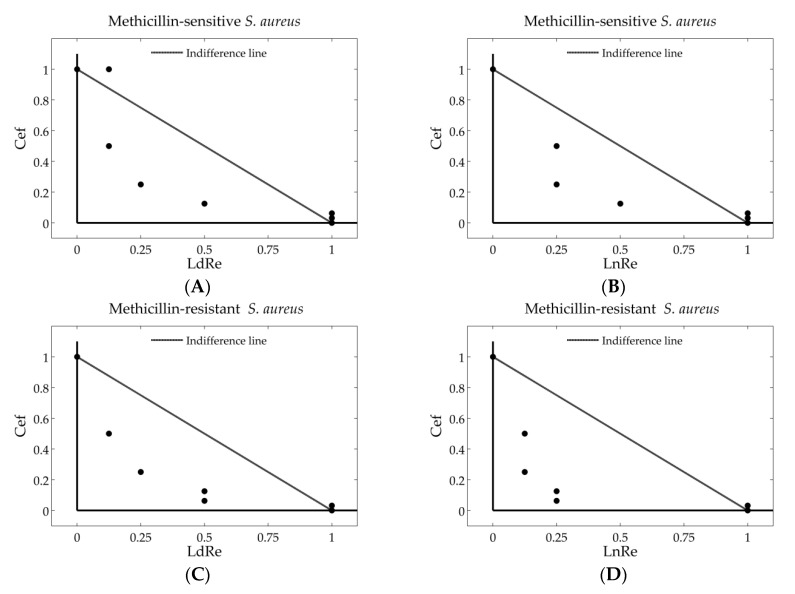
Normalized Isobolograms of LdRe or LnRe with cefotaxime against Methicillin-sensitive *Staphylococcus aureus* ATCC 25923 (MSSA) (**A**,**B**) and normalized Isobolograms of LdRe or LnRe with cefotaxime against Methicilin- resistant *S. aureus* ATCC 43300 (MRSA), respectively (**C**,**D**). The discontinuous line represents the indifference line and the points the experimental combinations at different levels.

**Table 1 antioxidants-10-00185-t001:** Ultrahigh resolution liquid chromatography orbitrap MS analysis (UHPLC-PDA-MS) orbitrap mass spectral data of LdRe and LnRe resins.

Peak	T_R_ (min.)	Tentative Identification	[M−H]^−^	UV Max nm	Theoretical Mass (*m*/*z*)	Measured Mass (*m*/*z*)	Accuracy (δppm)	MS_n_ ions (δppm)
1	1.40	Isosulochrin	C_17_H_15_O_7_^−^	278–337	331.08123	331.08212	2.69	146.96791
2	3.65	Axillarin	C_17_H_13_O_8_	2890–340	345.06049	345.06143	1.28	146.96791
3	4.10	Diosmetin	C_16_H_11_O_6_^−^	270–344	299.05501	299.05597	3.19	271.06097
4	4.26–4.30	meso-(rel 7S,8S,7′R,8′R)-3,4,3′,4′-tetrahydroxy-7,7′-epoxylignan	C_18_H_19_O_5_^−^	233–284	315.12380	315.12381		267.06481; 187.07582; 165.05512; 137.02363; 109.02862
5	4.80	Isorhamnetin	C_16_H_11_O_7_^−^		315.04993	315.05084	2.89	299.09216
6	5.00	3,4,3′,4′-tetrahydroxy 6,7′-Cyclolignan	C_18_H_19_O_4_^−^	233–284	299.12888	299.12872	3.13	241.05080; 190.92799
7	5.33	3 Methylluteolin	C_16_H_11_O_6_^−^	275–330	299.05501	299.05597	3.19	271.06097
8	5.65	Rhamnetin	C_16_H_11_O_7_^−^	256–367	315.04993	315.05099	3.38	299.09216
9	5.80	Tricin	C_17_H_13_O_7_^−^	256–367	329.06558	329.06644	2.60	285.0401
10	6.50	3,4,3′,4′-tetrahydroxy 6,7′-cyclolignan isomer	C_18_H_19_O_4_^−^	233–284	299.12888	299.12872	3.13	243.06482; 189.09164; 149.0600; 123.0443
11	6.80	NDGA isomer	C_18_H_19_O_4_^−^	228–283	301.14344	301.14435	3.03	174.95743; 122.03708
12	6.90	Norisoguaiacin isomer	C_19_H_21_O_4_^−^	231–286	313.14453	313.14432	2.81	247.09711
13	7.40	NDGA	C_18_H_21_O_4_^−^	228–283	301.14344	301.14429	3.01	174.95743; 154.02631; 137.02362; 122.03708
14	7.94	NDGA isomer	C_18_H_21_O_4_^−^	228–283	301.14344	301.14471	3.06	243.06614; 122.0367
15	8.42	Kaempferide	C_16_H_11_O_6_^−^	269–358	299.05501	299.05594	3.09	285.04053; 255.0279
16	8.61	Norisoguaiacin	C_19_H_21_O_4_^−^	231–286	313.14453	313.14461	3.01	301.1446; 241.0504; 173.06022
17	8.85	Trihydroxy-6,7′cyclolignan	C_19_H_18_O_3_^−^	233–284	283.13397	283.06111		268.03041; 240.04212; 211.03951; 117.03852
18	9.15	(7S,8S,7′R,8′R)-3,3′,4′-trihydroxy-4-methoxy-7,7′-epoxylignan	C_19_H_21_O_5_^−^	230–283	329.13945	329.17551	2.34	301.14410; 263.12848
19	9.37	4-[4-(4- hydroxy-phenyl)-2,3-dimethyl-butyl]-benzene-1,2-diol	C_18_H_21_O_3_^−^	282	285.14852	285.14948	3.0	154.02643; 137.02362; 122.03652
20	9.75	3′-MNDGA	C_19_H_23_O_4_^−^	230–283	315.15909	315. 15981	1.99	300.13622; 285.11272; 177.09142; 149.06005; 122.03653.
21	10.06	4′-MNDGA	C_19_H_23_O_4_^−^	230–283	315.15909	315.15991	1.62	300.13626; 285.11282; 177.09142; 149.06001; 122.03653
22	10.36	Lavandulilkaempferol	C_25_H_25_O_6_^−^	272	421.16456	421.16541	1.99	301.144134
23	11.04	Unknown NDGA derivative	C_27_H_29_O_6_^−^	280	449.19587	449.19659	1.62	301.144261
24	11.48	Unknown NDGA derivative	C_24_H_31_O_6_^−^	280	415.21152	415.21222	1.69	301.14425;
25	11.71	Unknown NDGA derivative	C_27_H_27_O_6_^−^	280	447.18022	447.18079	1.27	301.14423
26	11.81	Dihydroguaiaretic acid	C_20_H_25_O_4_^−^	230–283	329.17474	329.17557	2.52	301.14410; 263.12848
27	12.07	Dihydroguaiaretic acid 4′-O-acetate	C_22_H_27_O_5_^−^	283	371.18530	371.18597	1.81	301.14420
28	12.4	Reduced NDGA derivative	C_18_H_17_O_3_^−^	283	281.11722	281.11813	3.24	301.14420
29	12.85	Unknown NDGA derivative	C_25_H_25_O_5_^−^	283	405.16965	405.17147	2.02	301.14420
30	12.61	Dihydroguaiaretic acid 4″-O-acetate	C_22_H_27_O_5_^−^	276–352	371.18530	371.18595	1.73	301.14420
31	13.11	NDGA isomer	C_18_H_21_O_4_^−^	283	301.14344	301.14429	2.82	174.95712; 122.03708
32	13.37	3″,4″,4′-Trymethyl NDGA 3′-acetate	C_23_H_29_O_5_^−^	277	385.20195	385.20175	2.02	301.14423
33	13.68	NDGA isomer	C_18_H_21_O_4_^−^	280	301.14344	301.14429	2.82	174.95743; 122.03708
34	14.00	3″,4″,3′-Trymethyl NDGA 4′-acetate	C_23_H_29_O_5_^−^	283	385.20195	385.20172	2.00	301.14426; 122.03667
35	14.19	Unknown NDGA derivative	C_27_H_29_O_5_^−^	283	433.20095	433.20142	1.07	301.14429
36	14.56	Unknown NDGA derivative	C_26_H_33_O_6_^−^	283	441.22717	441.22781	1.46	301.14429
37	14.83	Unknown NDGA derivative	C_27_H_29_O_5_^−^	283	433.20095	433.20157	1.07	301.14426
38	15.08	3′MNDGA 3″, 4′-diacetate	C_24_H_31_O_5_^−^	283	399.21660	399.21738	1.94	301.14426
39	15.45	4′MNDGA 3′, 4″-diacetate	C_24_H_31_O_5_^−^	283	399.21660	399.21729	1.71	301.14426
40	15.68	3′MNDGA 4″, 4′-diacetate	C_24_H_31_O_5_^−^	283	399.21660	399.21738	1.94	301.14429
41	19.19	Unknown	C_24_H_23_O_3_^−^	-	359.16427	359.16513	2.67	235.92734

**Table 2 antioxidants-10-00185-t002:** Antioxidant properties and total phenolics and flavonoids content of LnRe and LdRe.

Assay	LnRe	LdRe
Content of phenols		
Total phenolics(mg GAE/g resin)	459.9 ± 10.07	390.46 ± 6.08
Flavonoids (mg QE/g resin)	40.8 ± 0.8	24.7 ± 1.8
Antioxidant		
DPPH (EC_50_ in µg resin/mL)	8.41 ± 0.04	8.42 ± 0.44
FRAP(mgTE/g resin)	1.94 ± 0.20	1.86 ± 0.16
TEAC (mgTE/g resin)	1.08 ± 0.07	1.09 ± 0.06
Percentage LP (at 100 µg resin/mL)	81.97 ± 0.11	81.03 ± 0.01

No significant differences were found between the three samples. ANOVA (analysis of variance) followed by Dunnett’s comparison test was used (significance *p* < 0.05).

**Table 3 antioxidants-10-00185-t003:** Nematicidal activity of LnRe and LdRe resins againstJ2 *M. incognita*.

		% Mortality Corrected		
Resins	Concentration			
		24 h	48 h	72 h
	1	3.49 ± 3.24 a	4.77 ± 3.89 a	10.44 ± 1.37 ab
LnRe	2	5.52 ± 3.54 a	11.77 ± 3.27 ab	17.76 ± 2.27 bc
	3	6.47 ± 4.42 ab	11.13 ± 3.15 ab	8.67 ± 2.73 a
	1	9.81 ± 1.45 ab	16.23 ± 1.86 bcd	24.03 ± 1.53 cde
LdRe	2	3.95 ± 3.83 a	13.98 ± 1.65 bc	28.56 ± 2.27 cde
	3	5.91 ± 4.69 a	15.61 ± 2.91 bc	20.76 ± 1.64 cd

Mean percent between rows at each column with different letters are significantly different (*p* = 0.05) according to LSD Fisher.

**Table 4 antioxidants-10-00185-t004:** Antibacterial activity of LdRe and LnRe resins (minimum inhibitory concentrations (MICs) and minimum bactericidal concentrations (MBCs) in µg resin/mL).

Bacterias	*Larrea* Resins				Reference Antibiotics			
	LnRe		LdRe		Cefotaxime		Imipecil	
Gram (+)	MIC	MBC	MIC	MBC	MIC	MBC	MIC	MBC
MSSA	16	32.5	16	32.5	0.5	0.5	0.5	0.5
MRSA	16	32.5	16	16	0.5	0.5	0.5	0.5
*Staphylococcus aureus*-MQ5097	250	250	250	250	0.8	1	1	1
*Streptococcuspyogenes*-MQ4	250	250	62.5	62.5	0.25	0.5	0,5	0.5
*Streptococcus agalactiae*	62.5	62.5	125	125	0.5	0.5	0,5	0.5
Gram (−)								
*Escherichiacoli* ATCC 25922	1000	1000	250	500	1.9	1.9	0,5	1
*E. coli*-MQ11009	400	500	250	250	1	1	0,5	1
*E. coli*-MQ11068	500	500	250	500	1	1	1	1
*E. coli*-MQ11062	250	500	500	500	1	1	1	1.5
*E. coli*-MQ586	62.5	125	62.5	125	1	1	1.5	1.5

MSSA: Methicillin-sensitive *Staphylococcus aureus* ATCC 25923; MRSA: Methicillin-resistant. *S. aureus* ATCC 43300; MIC: Minimum inhibitory concentration, MBC: Minimum Bactericidal Concentration.

**Table 5 antioxidants-10-00185-t005:** Combinatory effects of LdRe and LnRe with cefotaxime. The results are showed as MIC and MIC combination (MICcomb; expressed in μg/mL), fractional inhibitory concentration (FIC), fractional inhibitory concentration index (FICI) and dose reduction index (DRI) values.

	MSSA						MRSA					
	MIC	MICcomb	FIC	FICI	DRI	Effect	MIC	MICcomb	FIC	FICI	DRI	Effect
LdRe	32	16	0.5	1	2	Ind	32	32	1	1.25	4	Ind
Cef	1	0.5	0.5				0.5	0.125	0.25			
LnRe	32	32	1	1.25	4	Ind	16	8	0.5	1	2	Ind
Cef	1	0.25	0.25				0.5	0,25	0.5			

MSSA: Methicillin-sensitive *Staphylococcus aureus* ATCC 25923; MRSA: Methicillin-resistant *S. aureus* ATCC 43300; MIC: Minima Inhibitory Concentration; MICcomb: Minima Inhibitory Concentration combination; FIC: Fractional Inhibitory Concentration; FICI: Fractional Inhibitory Concentration Index; DRI: Dose Reduction Index; Ind: indifference. Synergy (FICI ≤ 0.5), no interaction or indifference (FICI > 0.5–4.0) and antagonism (FICI > 4).

## Data Availability

Not applicable.

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
