# Peer review of "UHPLC-ESI-OT-MS Phenolics Profiling, Free Radical Scavenging, Antibacterial and Nematicidal Activities of “Yellow-Brown Resins” from Larrea spp."

_antioxidants, 2021, doi:10.3390/antiox10020185_

Round 1

Reviewer 1 Report

Please write more data about the ethnobotanical background of these plants. It is collected, but when? Do we know anything about the variability of the active compounds dependent on the habitat, plant part, time of collecting, individual differences among the plants? 

Referring to the plant material please write properly which plant part was collected - in the text only "arial part" is mentioned. So, what was it? Flowering stems, stems with only leaves, from which part of the trees, in how many replications? How many plant did you choose for plant material collection, what was the amount, you collected? 

Author Response

Reviewer 1. Response

Manuscript ID antioxidants-1077979

Comments and Suggestions for Authors

Please write more data about the ethnobotanical background of these plants. It is collected, but when? Do we know anything about the variability of the active compounds dependent on the habitat, plant part, time of collecting, individual differences among the plants?

We appreciate your comments and suggestions which have been incorporated

The following paragraph was included in Introduction

The species of the genus Larrea in Argentina (Larrea ameghinoi, L. cuneifolia, L. divaricata and L.nitida), vernacular name "Jarillas" are used extensively in traditional medicine in Argentina and in Andean communities for the treatment of injuries and bruises, as well as disinfectant of wounds; repellent of insects, for roof construction in rural areas and as a vegetable fuel for cooking food. These uses are also shared with the resinous species Zuccagnia punctata, commonly called "Jarilla macho"[5]..

The following paragraph was included in section Results and discussion

In a previous work carried out in an Andean population in the center-west of Argentina, three marker compounds for L nitida were quantified over a year (February, May, September and November), highlighting their higher production in coincidence with the flowering period of the species, late spring and early summer (November and December)

Referring to the plant material please write properly which plant part was collected - in the text only "arial part" is mentioned. So, what was it? Flowering stems, stems with only leaves, from which part of the trees, in how many replications? How many plant did you choose for plant material collection, what was the amount, you collected?

The plant material paragraph was rewritten and the suggested information was included

Three representative and independent samples of 2 kg of aerial parts (stems with leaves and flowers) for each species were collected in December 2015, on Iglesia district, province of San Juan (Argentina) at an altitude of 1800 m (L. divaricata) and 2700 m (L. nitida) above sea level. Each sample was collected from ten plants

Reviewer 2 Report

Manuscript Antioxidants-1077979 entitled  " UHPLC-ESI-OT-MS phenolics profiling, free radical scavenging, antibacterial and nematicidal activities of “yellow-brown resins” from Larrea spp." by Jessica Gómez et al. is a research article and its aim is to investigate the potential antioxidant, antibacterial, and nematicidal of yellow-brown  resins extracted from plant species belonging to the genera Larrea.

The topic of this manuscript falls within the scope of Antioxidants.

The text is clear and easy to read.  The results are clearly presented. The methods are adequately described. The metabolite profiling and polyphenolic content was obtained by ultra-high resolution liquid chromatography orbitrap MS analysis (UHPLC-ESI-OT-MS), an outstanding and accurate technology to determine the presence of biologically active compounds in medicinal plants. The Authors used appropriate statistic methods. The conclusions are consistent with presented evidence and arguments. References are up to date and complete.

In my opinion this manuscript may be published after minor revision.

  • I suggest  removing the figure 1.
  • Table 2: Insert a space between "TEAC (mgTE / g resin) ..." and "" Percentage LP (at 100 μg resin / ml) ... "

Author Response

Reviewer 2. Response

Manuscript ID antioxidants-1077979

Reviewer 2

We appreciate your comments and suggestions which have been incorporated

Manuscript Antioxidants-1077979 entitled  " UHPLC-ESI-OT-MS phenolics profiling, free radical scavenging, antibacterial and nematicidal activities of “yellow-brown resins” from Larrea spp." by Jessica Gómez et al. is a research article and its aim is to investigate the potential antioxidant, antibacterial, and nematicidal of yellow-brown  resins extracted from plant species belonging to the genera Larrea.

The topic of this manuscript falls within the scope of Antioxidants.

The text is clear and easy to read.  The results are clearly presented. The methods are adequately described. The metabolite profiling and polyphenolic content was obtained by ultra-high resolution liquid chromatography orbitrap MS analysis (UHPLC-ESI-OT-MS), an outstanding and accurate technology to determine the presence of biologically active compounds in medicinal plants. The Authors used appropriate statistic methods. The conclusions are consistent with presented evidence and arguments. References are up to date and complete.

In my opinion this manuscript may be published after minor revision.

I suggest  removing the figure 1.

The Figure 1 was deleted from main text as suggested and moved to supplementary material because we consider that it could be of interest to potential readers of the journal (If you agree)

Table 2: Insert a space between "TEAC (mgTE / g resin) ..." and "" Percentage LP (at 100 μg resin / ml) ... "

Was inserted